# Molecular Docking and In Vitro Studies of Ochratoxin A (OTA) Biodetoxification Testing Three Endopeptidases

**DOI:** 10.3390/molecules28052019

**Published:** 2023-02-21

**Authors:** Pablo César Orozco-Cortés, Cesar Mateo Flores-Ortíz, Luis Barbo Hernández-Portilla, Josefina Vázquez Medrano, Olga Nelly Rodríguez-Peña

**Affiliations:** 1Laboratorio de Fisiología Vegetal, Unidad de Biología, Tecnología y Prototipos (UBIPRO), Facultad de Estudios Superiores Iztacala, Universidad Nacional Autónoma de México, Av. de los Barrios No. 1, Tlalnepantla 54090, Mexico; 2Laboratorio Nacional en Salud, Facultad de Estudios Superiores Iztacala, Universidad Nacional Autónoma de México, Av. de los Barrios No. 1, Tlalnepantla 54090, Mexico

**Keywords:** α–ochratoxin, *Ananas comosus* bromelain cysteine-protease, *Bacillus subtilis* neutral metalloendopeptidase, biodetoxification, bovine trypsin serine-protease, degradability, enzymatic mycotoxin detoxification, mycotoxin detoxifier

## Abstract

Ochratoxin A (OTA) is considered one of the main mycotoxins responsible for health problems and considerable economic losses in the feed industry. The aim was to study OTA’s detoxifying potential of commercial protease enzymes: (i) *Ananas comosus* bromelain cysteine-protease, (ii) bovine trypsin serine-protease and (iii) *Bacillus subtilis* neutral metalloendopeptidase. In silico studies were performed with reference ligands and T-2 toxin as control, and in vitro experiments. In silico study results showed that tested toxins interacted near the catalytic triad, similar to how the reference ligands behave in all tested proteases. Likewise, based on the proximity of the amino acids in the most stable poses, the chemical reaction mechanisms for the transformation of OTA were proposed. In vitro experiments showed that while bromelain reduced OTA’s concentration in 7.64% at pH 4.6; trypsin at 10.69% and the neutral metalloendopeptidase in 8.2%, 14.44%, 45.26% at pH 4.6, 5 and 7, respectively (*p* < 0.05). The less harmful α-ochratoxin was confirmed with trypsin and the metalloendopeptidase. This study is the first attempt to demonstrate that: (i) bromelain and trypsin can hydrolyse OTA in acidic pH conditions with low efficiency and (ii) the metalloendopeptidase was an effective OTA bio-detoxifier. This study confirmed α-ochratoxin as a final product of the enzymatic reactions in real-time practical information on OTA degradation rate, since in vitro experiments simulated the time that food spends in poultry intestines, as well as their natural pH and temperature conditions.

## 1. Introduction

About 100,000 fungi species have been identified, and more than 500 mycotoxins with toxigenic effects have been recognised [1] as a real public health issue that silently, still impacts, particularly in developing countries [2,3]. Mycotoxins are secondary metabolites produced by filamentous fungi, mainly of the Aspergillus, Fusarium and Penicillium genera [4] and have been found to infect fruits, grains, and seeds in their growth, harvest, drying and storage stages [5]. Their chemical structure along with the frequency of occurrence and the severity of the disease they produce, determines their importance and toxicity [6]. Throughout the more toxic mycotoxins, aflatoxins, fumonisins, zearalenone, certain ergot alkaloids, trichothecenes and ochratoxins, represented mainly by ochratoxin A (OTA), among others, have been considered [7,8]. Maximum levels have been set in European Union legislation to control these mycotoxin levels in food and feed [9]. Therefore, the search for effective, environmentally friendly and specific mycotoxin detoxifying methods are in great demand [10].

The modern direction of mycotoxin decontamination techniques is towards using non-invasive methods, with high specificity, with harmless products and that is environmentally friendly [11,12]. In this context, the use of enzymes that naturally occur in food commodities, produced during fermentation, or purified, have been found to detoxify mycotoxins by combining chemical and biological processing characteristics, besides high performance and specialisation and avoid causing toxicity to organisms [8,13]. Enzymatic biotransformation can degrade mycotoxins into non- or less-toxic metabolites [14,15,16]. Regarding the enzyme and mycotoxin types, mycotoxin biotransformation can be carried out by hydroxylation, oxide-reduction, hydrogenation, de-epoxidation, methylation, glycosylation and glucuronidation, esterification, hydrolysis, sulfation, demethylation, deamination and miscellaneous other chemical reactions [11,17]. Although several physical, chemical and biological strategies have been proposed to reduce or eliminate their levels in food and feed [18,19], enzymatic biological transformation has been considered one of the most promising, yet challenging approaches to degrade mycotoxins and reduce their accumulation [20]. However, to date, only a few enzymes have been identified, purified and characterised for this purpose [16,17,19,21].

OTA is a potent nephrotoxic, teratogenic, carcinogenic and immunotoxic, amongst others [22], and consists of an isocoumarin linked by its 7-carboxy group to L-β-phenylalanine by an amide bond [23]. OTA’s enzymatic detoxification could be achieved by the hydrolysis of either, (i) the amide bond to generate α-ochratoxin, and L-phenylalanine by using an amido-hydrolase, or (ii) the lactone ring by using an ochratoxin-lactonase [17,20,24]. Since OTA’s amide bond is similar to a peptide one, it is vulnerable to the action of hydrolytic proteases [13]. Proteases are widely distributed in a diversity of microorganisms, plants and animals, and endopeptidases, and their preferential action at the peptide bonds in the inner regions of the polypeptide chain away from the N and C termini, have been mainly classified into groups, based on the presence of specific amino acid substituents at their active sites, namely cysteine, serine and metallopeptidases [25]. To date, a limited number of purified serine [26] and metal-dependent carboxypeptidases [26,27,28,29,30,31,32,33,34,35] and ochratoxinases [24] have been proven to biotransform OTA [4,17,36,37,38,39,40,41] and references therein. There is variability in their efficiency of hydrolysing OTA, when regarding the enzyme identity and experimental conditions [28].

Seeking practical options of purified proteases as mycotoxin detoxifiers, due to their low cost, wide availability, ease of extraction and high resistance, the aim of this research, was to study the bromelain from *Ananas comosus* (cysteine), the bovine trypsin (serine), and a neutral metalloendopeptidase from *Bacillus subtilis* (metal-dependent) with potential to detoxify OTA through biotransformation. While bromelain is easily obtained from the stem and juice of pineapples, and is considered a very resistant enzyme [25], it has never before been evaluated for its ability to hydrolyse OTA; bovine trypsin, has been considered one of the most important pancreatic animal proteases, and as a digestive enzyme, hydrolyses peptide bonds where lysine and arginine residues contribute carboxyl groups, its study in the hydrolysis of OTA has only been addressed once in vitro at basic pH and room temperature conditions, where no OTA hydrolysis was found [26]; the neutral metalloendopeptidases require metal ions for their activity, and although some metal-dependent enzymes as carboxypeptidases (exopeptidase), have been recognised for hydrolysing OTA, the one studied in this study is a metalloendopeptidase. Analyses were performed using both, in vitro and in silico approaches. Molecular docking is an “in silico” technique currently used for the discovery of new drugs and treatments, and is capable of predicting the behaviour of small molecules when interacting in the active site of a protein, and can be applied to solve unambiguous issues in enzymatic structures to determine the catalytic mechanism [42,43,44]; and within the study of enzymes as mycotoxin detoxifying agents, allows visualisation of what is happening at the molecular level, however, studies including molecular approaches are still scarce [17], and there are even fewer of those dealing with the ability to hydrolyse OTA [17,24]. Moreover, unlike most previous studies, in which the considered incubation time was too long, such as hours [30,32,45], days [28,31,33,34,35] and even weeks [29,46], in this study, in vitro experiments mimicked the estimated time in which the food travels through the intestine (1 h), as well as considering the intestinal acidic pH conditions (4.5–5), and the chick’s natural body temperature (41 °C). The hypotheses are: (i) since bromelain is a protein-digesting enzyme that attacks the internal peptide bonds of the protein chain [47,48] it might be able to hydrolyse OTA’s peptide bond and bio-transform OTA into α-ochratoxin [11,49]; (ii) given that trypsin is a serine hydrolase that cleaves ester, amide, and thioester bonds in small molecules, peptides and proteins [50], it might be able to hydrolyse both, T-2 toxin by bio transforming it to HT-2 and OTA by opening its lactone ring [11] and (iii) due to the fact that neutral metalloendopeptidase is a metalloenzyme that hydrolyses peptide bonds [51,52,53], it might be able to hydrolyse OTA’s peptide bond and to bio transform OTA into α-ochratoxin [11,49]. Further, given the nature of the catalytic reaction carried out by bromelain and the neutral metalloendopeptidase, the T-2 toxin was used as a negative control for docking studies, since the epoxide ring and the two ester groups found in T-2 are not able to be broken.

## 2. Results

### 2.1. Validation

Ramachandran models of our tested proteins, bromelain and neutral metalloendopeptidase did not show residues in the outlier region.

### 2.2. Blind and Directed Docking

Results of blind molecular docking showed that the studied enzymes did not contain allosteric regions or ligand binding sites outside of those reported in previous papers (Table 1). The substrate binding pocket and the active site were found to be the most stable sites for T-2 and OTA binding. T-2 and OTA toxins showed a similar affinity for the substrate binding pocket as those of the reference ligands. Regarding the directed molecular docking, most of the poses that toxins adopted in the enzymatic active site were similar to those adopted by the reference substrates.

### 2.3. Multiple Sequence Alignment

Results showed that the three studied enzymes were distantly distributed and in different clusters, according to their genetic distance (Figure 1). The phylogenetic tree showed four main clusters, suggesting a different degradation mechanism of OTA. In the first one, the serine endopeptidases were grouped. The second cluster contained the serine-type carboxypeptidases. The third big cluster contained the metallocarboxypeptidases and bromelain, a cysteine endopeptidase, and although they had a different catalytic mechanism, they share 28% sequence homology. According to an exploratory analysis, structures mostly shared a similar orientation of alpha and beta sheets surrounding the active site (Figure 2). Further analysis must be performed to clarify these preliminary findings. The last cluster is only composed of the neutral metalloendopeptidase, highlighting its difference in form and catalytic mechanism of hydrolysing OTA.

### 2.4. Bromelain Cysteine Protease

#### 2.4.1. T-2 toxin

The T-2 toxin showed a higher free binding energy (ΔG), and its affinity (*K_i_*) for bromelain was approximately 5.44-fold lower than the reference ligand, indicating a higher affinity for the protein (E64; Table 2).

T-2 showed hydrogen bond (HB)-mediated interactions with amino acid residues of the ligand binding site (Table 3).

Although T-2 ester groups were located near the involved amino acids in the catalytic triad, given the nature of the catalytic reaction carried out by bromelain, the epoxide ring and the two ester groups found in T-2 are not going to be broken; instead, due to T-2′s high affinity (*K_i_* = 101.78 µM) and binding probability to form a complex (ΔG = −5.45 kcal/mol) it could function as a possible competitive inhibitor (Figure 3).

#### 2.4.2. OTA

OTA had, on average, more negative (ΔG) values when compared to T-2 and the reference ligand, showing that the binding is more spontaneous towards the bromelain binding site among all the evaluated ligands (Table 2). OTA had approximately 1.3-fold higher affinity when compared to the reference ligand than the T-2 toxin. OTA’s interactions with the bromelain binding site are mediated by a hydrogen bond (HB), aliphatic, and π-π T-shaped interactions with bromelain residues (Table 3). OTA’s amide group is oriented towards bromelain’s residues involved in the catalytic activity, Cys27, Gln20, His158 and Asn179 (Figure 4).

According to docking studies, the inactivation mechanism in OTA by bromelain triggers the catalytic action of Cys26 that attacks the carbonyl of the amide group in OTA, resulting in α-ochratoxin and β-phenylalanine as the final products (Figure 5).

Molecular dynamics showed that the bromelain-OTA complex maintained stable interactions (Table 4) within the active site until 50 (ns) when the complex started to dissociate. After 50 (ns), H-bond interactions started to diminish. The complex dissociated at 57 (ns) and OTA unbound from the enzyme (Figure 6).

Moreover, in vitro differences between groups were found (ANOVA *p* ≤ 0.0001; F = 16.87). Sidak’s multiple comparisons tests showed that bromelain reduced OTA concentration by 7.64 (%) at pH 4.6 (*p* ≤ 0.0001; t = 6.496; DF = 12; Figure 7). However, although the OTA ion was detected (404.08 *m*/*z*) the final product, α-ochratoxin, suggested in docking was not detected in chromatograms (212.85 *m*/*z*; Figure 8).

### 2.5. Bovine Trypsin Serine Protease

#### 2.5.1. T-2 Toxin

MXH reference ligand showed a higher free binding energy (ΔG) compared to T-2, and its affinity (*K_i_*) for trypsin was approximately 272-fold lower, indicating a higher affinity for the reference ligand (MXH; Table 2). T-2 toxin interactions are due to hydrogen bonds (HB), alkyl, and Van der Waals with trypsin residues (Table 3). T-2 is oriented towards trypsin’s residues involved in the catalytic activity, Asp102, His57 and Ser195 (Figure 9).

According to the docking studies, the inactivation mechanism in T-2 by trypsin triggers the catalytic action of Ser195 that attacks the carbonyl of the ester group in OTA, resulting in α-ochratoxin and β—phenylalanine as the final products (Figure 10).

Molecular dynamics showed that the trypsin–T2 complex maintained stable interactions (Table 4) within the active site until 60 (ns), when the complex started to dissociate. After this time, H-bond and Van der-Waals interactions diminish. The complex dissociated at 63 (ns) and T-2 unbound from the enzyme (Figure 11).

In vitro experiments testing trypsin’s capacity to hydrolyse T-2 toxin, showed no significant changes when comparing experimental samples at different pH conditions (ANOVA *p* = 0.07; F = 2.748; Figure 12); however, Sidak’s multiple comparisons found that the average percentage of the three repetitions at pH 4.6 had a significant decrease in T-2 concentration in 10.68% (*p* = 0.04; t = 2.86; DF = 12). Further, when HPLC–TOF–MS chromatograms were analysed, Both, T-2 and HT-2 corresponding ions were present, 484.25 (*m*/*z*) and 442.24 (*m*/*z*), for T-2 and HT-2, respectively (Figure 13). Since this study is mainly related to OTA hydrolysis, the related T-2 hydrolysis by trypsin is not discussed, due to the marginal results obtained in this study, further suggesting that extensive studies are required in this regard.

#### 2.5.2. OTA

MXH reference ligand showed a higher free binding energy (ΔG) compared to OTA, and its affinity (*K_i_*) for trypsin was approximately 94-fold lower, indicating a higher affinity for the reference ligand (MXH; Table 2). OTA molecular interactions are mediated by hydrogen bonds (HB), amide π, and Van der Waals interactions with trypsin residues (Table 3). OTA’s amide group is oriented towards trypsin’s residues involved in the catalytic activity, Ser195, His57 and Asp102 (Figure 14).

According to docking studies, the inactivation mechanism in OTA by trypsin triggers the catalytic action of Ser195 that attacks the carbonyl of the amide group in OTA, resulting in α-ochratoxin and β—phenylalanine as the final products (Figure 15).

Molecular dynamics showed that the trypsin–OTA complex maintained stable interactions throughout the 100 (ns; Table 4), despite that OTA had movement on its own axis, it always remained near trypsin’s active site and the complex never dissociated (Figure 16).

Moreover, in vitro differences between groups were found (ANOVA *p* ≤ 0.0001; F = 1.51). Sidak’s multiple comparisons tests showed that bromelain reduced OTA concentration by 10.69 (%) at pH 4.6 (*p* ≤ 0.0001; t = 8.88; DF = 12; Figure 17). The final product was identified unequivocally as α-ochratoxin (Figure 18).

### 2.6. Neutral Metalloendopeptidase

#### 2.6.1. T-2 Toxin

T-2 toxin showed a similar free binding energy (ΔG) compared to those of the reference ligand (DB0763; Table 2). However, its affinity (*K_i_*) for neutral metalloendopeptidase was approximately 17-fold higher than the reference ligand, indicating a very low affinity for the protein. T-2 showed hydrogen bond (HB), aliphatic and π–σ–mediated interactions with amino acid residues of the ligand binding site (Table 3). Although T-2 ester groups were located near the involved amino acids in the catalytic triad, given the nature of the catalytic reaction carried out by neutral metalloendopeptidase, the epoxide ring and the two ester groups found in T-2 are not going to be broken. Instead, it could function as a possible competitive inhibitor (Figure 19).

#### 2.6.2. OTA

OTA’s most negative value (ΔG) value was -8.56, which is very similar to that of the reference ligand (DB07673) when both were calculated by Swiss Dock, which is considered the most precise software when metallic ions are present (Table 2). Reference ligand showed 27.7-fold higher affinity (*K_i_*) when compared to OTA and neutral metalloendopeptidase.

OTA’s interactions with neutral metalloendopeptidase binding site residues are mediated by a hydrogen bond (HB) and aliphatic interactions (Table 3). OTA’s peptide bond is oriented towards neutral metalloendopeptidase residues involved in the catalytic activity, His373, His369, His453 and Glu393 (Figure 20).

The inactivation mechanism in OTA by neutral metalloendopeptidase triggers the catalytic action of the Zn that attacks the carbonyl of the amide group in OTA, resulting in α-ochratoxin and β-phenylalanine as final products (Figure 21).

Molecular dynamics showed that metalloendopeptidase–OTA complex maintained stable interactions throughout the 100 (ns; Table 4), despite OTA slightly moving from the catalytic triad, but remaining within the ligand binding site and the complex never dissociated (Figure 22).

In vitro experiments showed differences in the neutral metalloendopeptidase capacity to reduce OTA at different pH conditions (ANOVA *p* ≤ 0.0001; F = 464; Figure 23). Sidak’s multiple comparisons tests showed differences between the control and all tested pH’s treatments, at pH 4.6 in 8.2% (*p* ≤ 0.0001; t = 7.07; DF = 12); at pH 5 in 14.44% (*p* ≤ 0.0001; t = 12.47; DF = 12); at pH 7 in 45.26% (*p* ≤ 0.0001; t = 39.08; DF 12). The final product was identified unequivocally as α-ochratoxin (Figure 24).

## 3. Discussion

Biological detoxification using enzymes extracted from natural products, or produced by microorganisms, seems to be one of the most promising approaches to reduce or completely remove mycotoxin contamination from food [40]. It is especially worth trying to obtain inexpensive, easy-to-extract and resistant enzymes. In this study, the original predictions were that the proteases, bromelain cysteine from *Ananas comosus*, bovine trypsin serine and the neutral metalloendopeptidase from *Bacillus sutbtilis* would biodetoxify OTA by hydrolysing its peptide bond, according to the shared characteristics between the studied enzymes and those previously reported OTA degrading enzymes, including the type of reaction carried out by the enzyme, functional parameters, molecular properties, sequence and structural homology (See Figure 1). To our knowledge this is the first attempt to demonstrate that: (i) bromelain and trypsin are capable of hydrolysing OTA at acidic pH conditions but with low effectiveness; and (ii) the *Bacillus sutbtilis* neutral metalloendopeptidase was demonstrated to have a high efficacy as an OTA biodetoxifier. Moreover, in all the studied enzymes, both results obtained by docking and in vitro, were consistent, although in vitro experiments showed differences in enzymatic efficiency to hydrolyse OTA. Besides, α-ochratoxin was confirmed as a final product of the enzymatic reaction with trypsin and the metalloendopeptidase due to OTA’s degradation and not due to adsorption. These results are of practical use because pH conditions and contact time are comparable to the digestion process of poultry, constituting an important contribution of this study. This section will discuss the most important findings and their implications are discussed.

This study is the first attempt at evaluating the capacity of bromelain for OTA detoxification, despite being an easy-to-obtain, low-cost and highly resistant enzyme. Although findings by docking showed it was likely that OTA might be detoxified by bromelain, and despite experiments being performed at three different pH levels, results were only significant at the most acidic pH (4.6); however, hydrolysis per cent was low. Since bromelain is a cysteine/thiol-type enzyme containing a Cys–His catalytic dyad, this group of proteases is generally cut from the left side [59]. Thus, catalysis is unlikely to occur if OTA couples backwards to the active site. Moreover, as a protease, bromelain acts specifically, and the peptide bond of Arg–X is reported as its preferable cleavage site [60]. It also needs to recognise the Arg side chain. Since OTA structures do not have these similarities with Arg, it is, therefore, possible that the enzyme may not recognise the cleavage site efficiently. Results showed that although OTA reduced its concentration in the presence of bromelain, there was a lack of detection of the α-ochratoxin ion. This could be because the bromelain degradation rate might be very low and does not accumulate enough α-ochratoxin to be detected. However, further studies are needed to better clarify these assertations including a wide range of pH’s, temperatures, incubation times, substrate concentrations, and other cofactors. Moreover, molecular dynamics studies showed that the OTA–bromelain complex had stability until 50 (ns), suggesting that interactions in the binding site are strong enough to keep the ligand near to the catalytic triad during the time required to hydrolyse OTA. Further studies are needed to delve into this matter.

The results demonstrated that trypsin was effective in hydrolysing OTA at pH 4.6. Since chick’s intestinal pH is acidic, our results suggest that trypsin may be able to hydrolyse OTA in vivo, although OTA’s degradation percentage was below 20%. The only previous study in which OTA hydrolysis using trypsin was evaluated at pH 7.5 and 25 °C, found no OTA hydrolysis [26]. Future studies using trypsin in OTA biotransformation should develop in vivo assays. Further, molecular dynamics demonstrated that OTA was always next to the trypsin active site. This result suggest that the protein–ligand complex is therefore stable and its low degradation rate as demonstrated in in vitro experiments may be due either to non-optimum experimental conditions, or due to OTA being a long time near the active site, as shown in the dynamics, it could be hydrolysed and the product could be behaving as an inhibitor if it has a high affinity for the trypsin active site. However, this statement requires further investigation.

Results showed that the neutral metalloendopeptidase from *Bacillus sutbtillis* was able to degrade OTA in acidic and neutral pH conditions. Similar values have been previously reported for other metallopeptidases such as carboxypeptidases, from *Bacillus amyloliquefaciens* [31] and from Aspergillus niger [36] where OTA was found to be degraded by 72 and 99%, respectively, and others [26,27]. Alternatively, the similarity of results found in this study between the pH conditions 4.6 and 5, can be explained in terms of OTA’s ionisation value (4.4 pKa) [61], in which, regardless of OTA’s carboxyl ionisation state, non-significant differences in the mycotoxin transformation were observed. In fact, in the proposed degradation mechanism of OTA, the carboxyl does not take part in the hydrolysis/ionisation. Moreover, although the neutral metalloendopeptidase from *Bacillus sutbtilis* in this study, degraded OTA at pH 5, it was greater at pH 7, which agrees with those found for a carboxypeptidase by Abrunhosa et al. (2006) [36]. This study reported as optimal pH conditions, values between 6.5 and 7.5. However, they did not obtain OTA degradation testing this enzyme at 50 (°C). Results showed that the metalloendopeptidase effectively degrades OTA at 41 (°C). Further, this is the first attempt to report a metalloendopeptidase with the ability to hydrolyse OTA. According to molecular dynamics results, the metalloendopeptidase–OTA complex stays stable over time, what matches with the high amount of OTA degradation, obtained in vitro.

Results showed that the final OTA metabolite was α-ochratoxin as a product of the reaction in the three tested enzymes. OTA’s enzymatic detoxification could be achieved by the hydrolysis of either: (i) the amide bond to generate α-ochratoxin, and L-phenylalanine by using an amido-hydrolase, or (ii) the lactone ring by using an ochratoxin-lactonase [17,19,24]. However, when compared with the lactone ring opening, the hydrolysis of the amide bond between the phenylalanine and α-ochratoxin, as obtained in this study, has been widely recognised as essentially non-toxic products. α-ochratoxin is the isocoumarin section of OTA. Transforming OTA into α-ochratoxin is an efficient way to reduce not only its concentration but its toxicity [40], since α-ochratoxin has been described as the less toxic member of ochratoxins, where OTA, the most toxic is followed by OTC, OTB and finally by α-ochratoxin [62,63,64,65]. In fact, α-ochratoxin has been found to be 1000-times less toxic in brain cell cultures and its elimination half-life in vivo to be ten-times faster than for OTA [36], and to date, has been further chemically identified, as an OTA degradation product by microorganisms and enzymes [41].

Further, the (2S)-2-methyl-3-phenylpropanoic acid (DB07673), the chosen reference ligand for the neutral metalloendopeptidase protease docking, had the highest affinity (*K_i_*) when compared with the tested ligands (T-2 and OTA). This molecule is an inhibitor by nature, due to its relatively small size, its phenyl and carboxylic acid groups, and the fact that at pH 7 it gets deprotonated. Such traits allow it to easily enter the catalytic site, where its deprotonated carboxylic acid group can perfectly coordinate with the zinc metal ion [66]. Other metallopeptidase inhibitors have also observed the same behaviour [67,68,69,70].

Results showed that trypsin hydrolyses T-2 toxin, even though it was with a low efficiency. However, molecular docking studies showed that T-2 is near the catalytic triad, and molecular dynamics results showed that the trypsin–T2 complex was stable, on the whole, further suggesting that optimal experimental conditions have not yet been reached for a higher hydrolysis yield. Moreover, considering that two out of the three enzymes studied, do not break ester bonds, the T-2 toxin was used as a negative control. However, given its arrangement within both, bromelain and neutral metalloendopeptidase binding Sites, as found in docking, it is therefore possible that the toxin could act as a natural inhibitor. If this statement were true, the amount of enzyme could be increased so it can be inhibited by T-2 leaving the rest free so it might be available to act with OTA. Further, if T-2 is interacting with the enzyme, then although the enzyme cannot degrade it, it would be limited and could not be adsorbed in the animal’s intestine. Since mycotoxins often co-occur in food and feed, it has been shown that they can act harmfully due to interactions between them [71,72]; a stronger synergistic effect has been found for mixtures containing T-2 toxin and OTA [73]. Therefore, results obtained in this study further highlight the importance of continuing to seek effective, environmentally and specific enzymatic options that can detoxify not only one, but several mycotoxin types.

## 4. Materials and Methods

### 4.1. In Silico

#### 4.1.1. Protein–Ligand Preparation

The protein structure of the cysteine protease from Ananas comosus (bromelain; PDB ID: 6YCF) and the serine protease from bovine (trypsin; 6T9V) were obtained from the Protein Data Bank^®^ (https://www.rcsb.org/) and chain A was selected. Neutral metalloendopeptidase B (nprB) was modelled by homology using as template, neutral metalloendopeptidase C (Bacillolysin) from Bacillus cereus (PDB ID: 1NPC). Filling missing amino acids and homology modelling were performed in SwissModel^®^ (https://swissmodel.expasy.org/; 12 July 2022). Homology modelling were performed using enzymes own sequence (Uniprot ID: Q7DNA3 and P39899; for bromelain and neutral metalloendopeptidase, respectively). Energy minimisation was performed using Chimera 1.16^®^ (https://www.rbvi.ucsf.edu/chimera) [74]. Ramachandran analyses were performed to validate enzymes’ geometry using MolProbity^®^ (http://molprobity.biochem.duke.edu/index.php [75]. Structures of tested ligands T-2 toxin (PubChem ID: 5284461) and ochratoxin A (Pubchem ID: 442530), and those of reference ligands E64 (Drugbank ID: DB04276), MXH (Pubchem: 154815568) and diamino-methyl-phenylalanine (Drugbank ID: DB07673) were drawn using ChemSketch 2021.2.0^®^ (www.acdlabs.com; 3 August 2022; Table 1). Ligand geometry was optimised at the molecular mechanics level (AM1 basis set) by using Gaussian 09 software^®^ [76]. Multiple sequence analyses (MSA) were performed to compare the sequence homology of our studied enzymes and those of others previously reported [77] using Clustal Omega version 1.2.4 (https://www.ebi.ac.uk/Tools/msa/clustalo/, 11 December 2022) and Unipro UGENE v45.0 [78].

#### 4.1.2. Molecular Docking Studies

Analyses were performed in Autodock Vina [79,80], Swiss Dock [81] and Autodock 4.2.6 [59] due to differences in their molecular docking software’s scoring function, which reduces result biases and increases reliability. To prepare ligand and protein docking files for Autodock Vina and Autodock 4.2.6, Auto Dock Tools 1.5.6 was used [82]. In brief, the remaining water, ligands and ion molecules were removed prior to adding polar hydrogens and Kollmann charges to the protein structures. Gasteiger charges were calculated for each ligand. Finally, Autodock 4.2.6. output files were used in SwissDock.

Molecular docking was performed by using blind and directed approaches. Blind docking was used to explore binding sites in the whole crystal to identify or discard putative allosteric binding sites using a grid box of 120 × 120 × 120 Å. Directed docking was used to evaluate ligand interactions directly in the ligand binding site of each protein, and a grid-box of 60 × 60 × 60 Å was centred on the site where their reference ligand was removed from the crystallised protein. While a Lamarckian genetic algorithm with an initial population of 100 random individuals and 1 × 10^7^ iterations was used for Autodock Vina and AutoDock 4.2. [83], an “accurate” docking type was selected in SwissDock. Results were analysed for affinity estimation values and interactions with AutoDockTools 1.5.6 [84]. Ligand affinity for the protein was evaluated by calculating binding free energy values (ΔG) and were calculated in Autodock Vina, SwissDock and Autodock4, where the more negative the value, the more spontaneous is the binding between ligand–protein. The theoretical value of inhibition (*K_i_*) was calculated in Autodock4, where smaller values indicate a higher ligand affinity for the protein. Images were performed in Discovery Studio^®^ [85] and PyMOL^®^ [86]. To validate the docking procedure, the re-docking of the removed ligand on the enzyme was performed and evaluated.

#### 4.1.3. Molecular Dynamics

To perform the equilibrium molecular dynamics (EMD), files from Autodock 4 tool ligand–protein complexes (.pdb) were prepared using Visual Molecular Dynamics software (VMD v.1.9.4). A water box 10 Å spaced from the protein edge was constructed. The potential CHARMM36m was used to represent the molecules’ force field [87], and the system was neutralized using NaCl ions (0.15 mM). A Langevin dynamic was established to maintain 310 K (36.85 °C). While in the minimization process 2500 steps were used, for the equilibration process (NVT) 2,500,000 steps were (5 ns), and for the final simulation time (NPT) 50,000,000 steps 100 (ns). The time step was set to 2 fs/step. EMD were performed using the NAMD 2.14 software [88]. Trajectories were analysed by calculating the mean root square deviation (RMSD) with the RMSD plugin using the Visualizer Tool of VMD 1.9.4. Finally, trajectories were plotted using the Xmgrace software.

### 4.2. In Vitro Experiments

Control tubes consisted of pure mycotoxin standard 100 (µg/Kg) and phosphate buffer with a final volume of 1 (mL). The OTA detection limit was 14.11 (µg/L), and the quantification limit 23.45 (µg/L). Tested tubes, contained a mixture with a final volume of 1 (mL) containing 20 (mg) of the enzymes, either bromelain from *Ananas comosus* (Sigma^®^, Indonesia), trypsin from bovine (Sigma^®^), or neutral metalloendopeptidase from *Bacillus subtilis* (Neutroma^®^, JiangXi, China), OTA pure standard 100 (ppb; Trilogy TSL-504^®^, Washington, DC, USA) and 150 (mM) phosphate buffer. For all tested enzymes, assays were performed at pH 4.6, 5 and 7, at 41 (°C) and the incubation period lasted 60 (min). Tubes were centrifuged at 18,000× *g* for 2 (min). T-2 (1000 µg/Kg) was only tested with trypsin under the same experimental conditions as OTA.

#### 4.2.1. Residual OTA Analysis

Initial OTA quantification was assessed using a reverse phase with an HPLC (Agilent Technology 1100 series) coupled to a fluorescence detector (Perkin Elmer, Beaconsfield, Buckinghamshire, UK; LS50B), at an excitation wavelength of 345 (nm) and an emission wavelength of 455 (nm). A column DISCOVERY C-18 (250 × 4.6 mm; 5-micron, Supelco, USA) was used at a temperature of 25 (°C). The mobile phase consisted of acetonitrile: 0.25 N phosphoric acid (50:50). The run was isocratic with a constant flow of 1 (mL/min). Data collection was done by FL Winlab (version 2.0; The Perkin-Elmer Corporation). Samples were injected in triplicate and chromatograms were compared with those of their own control.

To identify final metabolites, when the tested enzyme significantly reduced OTA in samples compared with their own control, a reverse phase HPLC–ESI–TOF–MS (High Pressure Liquid Chromatography–Electrospray-Time of Flight–Mass spectrometry) was used, in positive ion mode, using an Agilent 1260 Infinity HPLC system equipped with a C-18 column (2.1 × 100 mm; 1.8 micron, ZORBAX Eclipse Plus, CA, USA); 1.8 Micron (Agilent Technologies, CA, USA) at a temperature of 25 (°C). The mobile phase consisted of water: formic acid (1%) and acetonitrile: formic acid (1%) with a flow rate of 0.15 (mL/min) and an injection volume of 40 (µL). The HPLC was coupled to a TOF/MS (Agilent 66230B) with an electrospray interface [89]. Gas temperature was 350 (°C), gas flow 6 (L/min) and nebuliser pressure 50 (psig), shredder 100 (V), skimmer 65 (V) and OCT RF vpp 750 (V), capillary voltage 3500 (V). Final precursor ions were identified at 404.05 and 212.84 (*m*/*z*), OTA and α-ochratoxin, respectively. The total range examined was from 100 to 1000 (*m*/*z*). Samples were injected in triplicate and data were registered with the acquisition software Mass Hunter Workstation LC/MS Data Acquisition version B.05.01.

#### 4.2.2. Residual T-2 Analysis

T-2 quantification was carried out by reverse phase HPLC–ESI–TOF–MS, in a positive ion mode using an Agilent 1260 Infinity HPLC (Agilent Technologies, Yishun, Singapore) system equipped with an RRHD Eclipse Plus C-18 column (1.8 µm, 2.1 × 100 mm; Agilent Technologies, USA); at 25 (°C). The mobile phase consisted of (A) ammonium acetate (10 mM) and (B) HPLC-grade methanol. The gradient started with 80 (%) A and 20 (%) B. Flow rate was set to 0.15 (mL/min) and 40 (µL) for the sample injection volume. The HPLC was coupled to an Agilent 66230B TOF/MS with an electrospray interface. The gas temperature was at 350 (°C), gas flow 6 (L/min) and nebuliser pressure 50 (psig), shredder 105 (V), skimmer 60 (V) and Oct RF 750 (V), capillary voltage 4000 (V). The final precursor ion for T-2 was identified at 484.25 (*m*/*z*) and 442.24 (*m*/*z*) for HT-2. The total range examined was from 100 to 1000 (*m*/*z*). Samples were injected in triplicate. Data were analysed in the Mass Hunter Data Acquisition software for 6200 series (version 5.01.5125; Agilent Technologies, USA) and Qualitative Analysis (version 6.0.633.10 (Aligent Technologies, 2017).

#### 4.2.3. Data Analysis

To determine whether differences in OTA and T-2 concentrations (µg/L) exist between control and tested tubes under different pH and temperature conditions, one-way multiple analysis of variance (ANOVA) was performed, followed by Sidak’s post hoc tests [90,91]. A *p* < 0.05 was considered significant. Analyses and figures were performed using Prism 8^®^ Version 8.4.0 for Mac OS, (GraphPad Software, Inc., La Jolla, CA 92037, USA).

## 5. Conclusions

Overall, this study provides consistent in silico and in vitro evidence, that bromelain and trypsin are capable of hydrolysing OTA in acidic pH conditions at a low efficiency; The *Bacillus sutbtilis* neutral metalloendopeptidase has a great effectiveness as an OTA bio detoxifier, confirming α-ochratoxin as a final product of the enzymatic reactions and providing real-time practical information on OTA degradation rate.

## Figures and Tables

**Figure 1 molecules-28-02019-f001:**
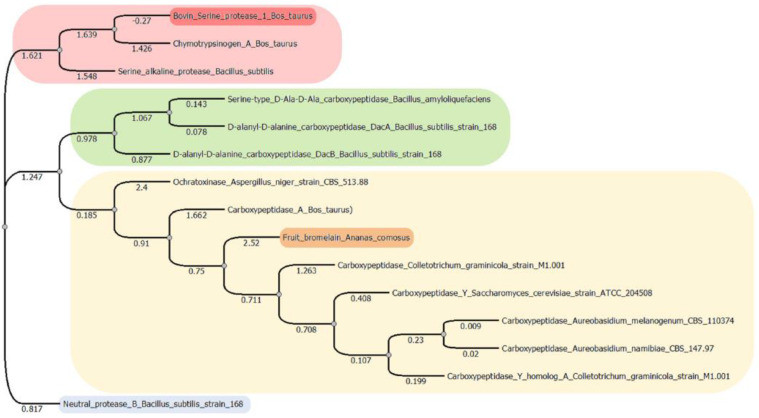
Phylogenetic tree showing genetic distance distribution of OTA hydrolysing enzymes (original figure). The tree was constructed using multiple sequence alignment (MSA). Sequences belong to the three studied enzymes in this study: bovine serine protease (highlighted in red), fruit bromelain (*Ananas comosus*; highlighted in orange) and neutral metalloendopeptidase (*Bacillus subtillis*; highlighted in blue); and those taken from previous studies found to bio transform OTA: chymotrypsin (*Bos taurus* [26]); serine alkaline protease (*Bacillus subtillis* [28]); Serine type D-Ala-D-Ala carboxypeptidase (*Bacillus amyloliquefaciens* [31]); D-alanyl-D-alanine carboxypeptidase DacA strain 168 (*Bacillus subtillis* [35]); D-alanyl-D-alanine carboxypeptidase DacB strain 168 (*Bacillus subtillis* [35]); ochratoxinase strain CBS513.88 (*Aspergillus niger* [24]); carboxypeptidase A (*Bos taurus* [21,26]); carboxypeptidase strain M1.001 (*Coletotrichum graminicola* [57]); carboxypeptidase Y strain ATCC204508 (*Saccharomyces cerevisiae* [57]); carboxypeptidase CBS110374 (*Aureobasidium melanogneum* [32]); carboxypeptidase CBS14797 (*Aureobasidium namibidae* [58]); carboxypeptidase Y homolog A strain M1.001 (*Colletotrichum graminicola* [57]).

**Figure 2 molecules-28-02019-f002:**
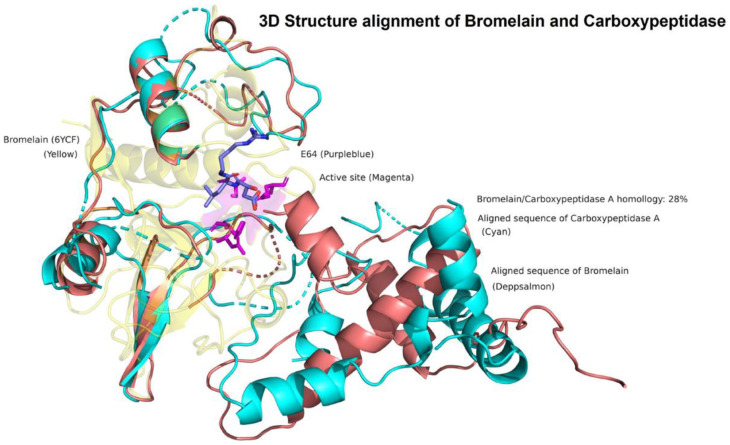
3D Structure alignment of Bromelain and Carboxipeptidase A. The exploratory analysis showed that Bromelain and Carboxypeptidase A mostly share a similar structural conformation surrounding the binding pocket, despite its 28% sequence homology and different catalytic mechanisms. Bromelain (yellow); Carboxypeptidase A aligned sequence (cyan); Bromelain aligned sequence (deep salmon); bromelain active site residues (magenta); E64 protease inhibitor (purple–blue).

**Figure 3 molecules-28-02019-f003:**
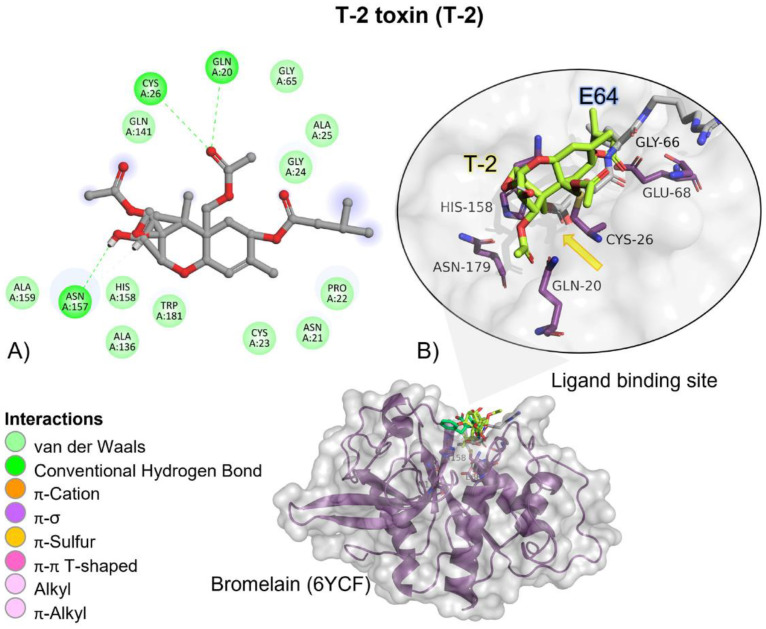
T-2 toxin—bromelain interactions. (**A**) Detailed view showing the interactions of the main different amino acid residues within the binding pocket of bromelain with the T-2 toxin ligand. (**B**) Panoramic view of the reference (E64) and the tested T-2 toxin ligands within the binding pocket of bromelain. Comparison between the docked pose of the tested and the reference ligand within the bromelain binding pocket. The yellow arrow indicates the proximity between the bromelain catalytic triad and the T-2 toxin.

**Figure 4 molecules-28-02019-f004:**
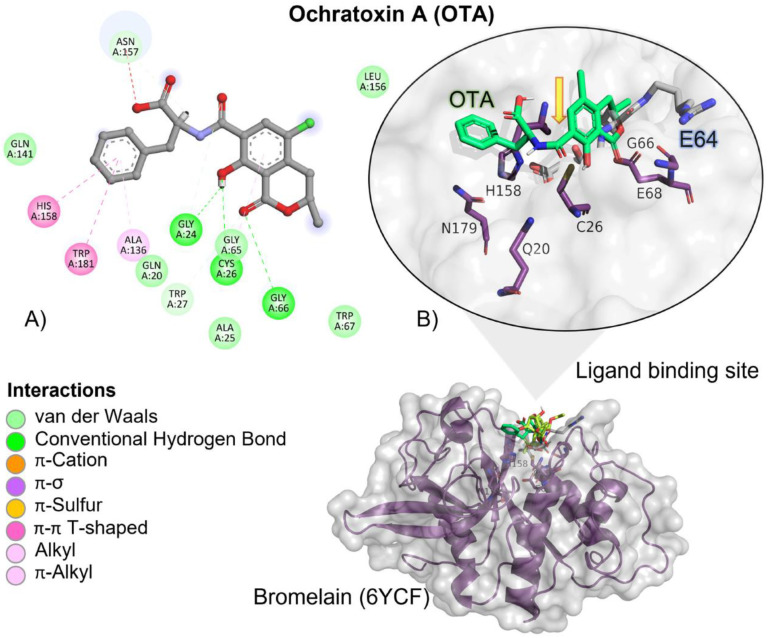
Ochratoxin (OTA)—bromelain interactions. (**A**) Detailed view showing the interactions of the different amino acid residues within the binding pocket of bromelain with the OTA ligand. (**B**) Panoramic view of the reference (E64) and the tested OTA ligands within the binding pocket of bromelain. Comparison between the docked pose of the tested and the reference ligand within the bromelain binding pocket. The yellow arrow indicates the proximity between the bromelain catalytic triad and the OTA amide bond, suggesting its breakup.

**Figure 5 molecules-28-02019-f005:**
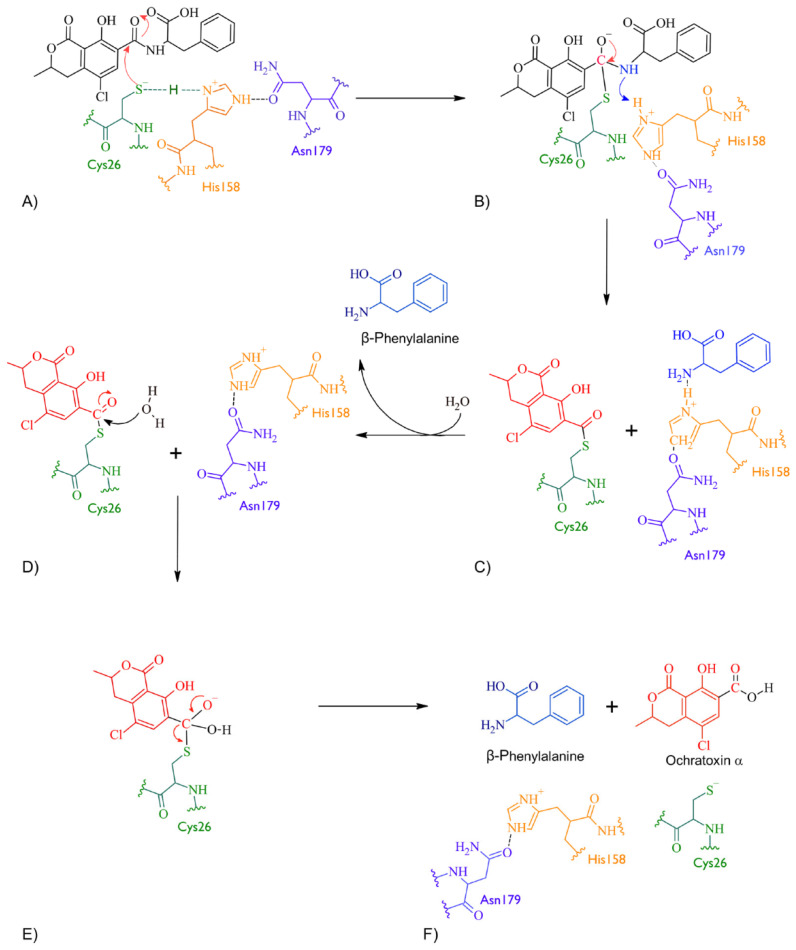
The detoxifying mechanism of OTA by bromelain. (**A**) Cys26 in bromelain attacks the carbonyl in OTA. Cys26 in its thiolate form and the imidazole side chain of His158 is protonated and therefore positively charged, forming an ion pair. (**B**) An intermediate is formed between Cys26 in bromelain and OTA. The electron pair is restored and the double bond between the carbonyl and oxygen is re-established. The amino group in the amide interacts with Hys158 and the amide bond breaks. (**C**) An intermediate is formed between Cys26 and the OTA fragment containing the carbonyl and another intermediate between Hys158 and OTA’s amino-containing fragment. (**D**) The β-phenylalanine product is released and the Hys158 restored. A water molecule is used to hydrolyse the intermediate between Cys26 and OTA’s carbonyl-containing fragment. The oxygen in the water molecule attacks the carbonyl and delocalises an electron pair to the oxygen of the carbonyl. (**E**) The carbonyl double bond is restored, and the bond formed between Cys26 and OTA’s carbonyl-containing fragment is broken. (**F**) The amino acids at the bromelain binding site are restored and α-ochratoxin product is released.

**Figure 6 molecules-28-02019-f006:**
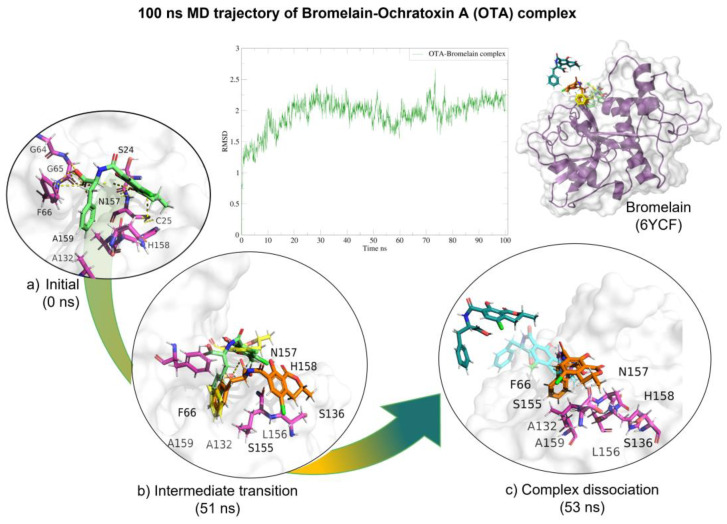
Molecular dynamics of the bromelain-OTA complex. The root mean squared deviation (RMSD) plot shows changes in the complex conformation with respect to the initial state during 100 (ns). (**a**) Initial pose (0 ns), complex initial state obtained from molecular docking studies. (**b**) Intermediate transition poses visualization where OTA moves within the ligand binding site. (**c**) Complex dissociation, OTA unbound from bromelain binding site. Green, OTA’s initial pose; Yellow, OTA at 31 (ns); Orange, OTA at 51 (ns); Blue, OTA at 52 (ns), Dark green, OTA at 53 (ns).

**Figure 7 molecules-28-02019-f007:**
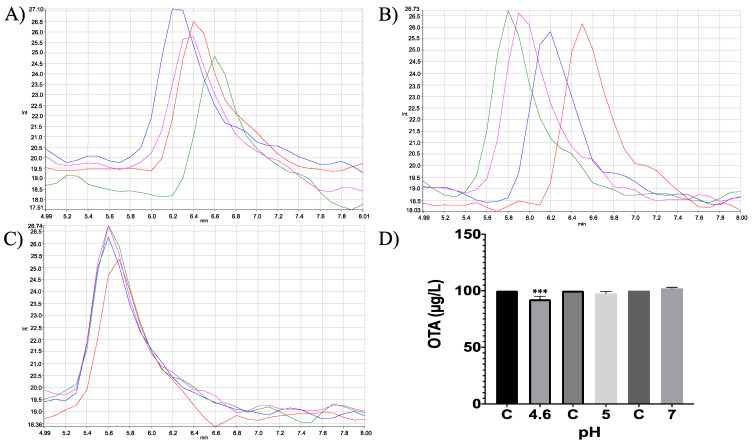
HPLC- fluorescence chromatograms of bromelain -OTA in vitro experiments showing differences in bromelain’s capacity to hydrolyse OTA at different pH conditions and incubated at 41 °C. (**A**) pH 4.6; (**B**). pH 5; (**C**) pH 7; (**D**) Bars show OTA (µg/L) degradation (7.64%) by bromelain at pH 4.6. In all chromatograms control is represented in red and experimental repetitions are in blue, green and pink. (*** *p* < 0.0001).

**Figure 8 molecules-28-02019-f008:**
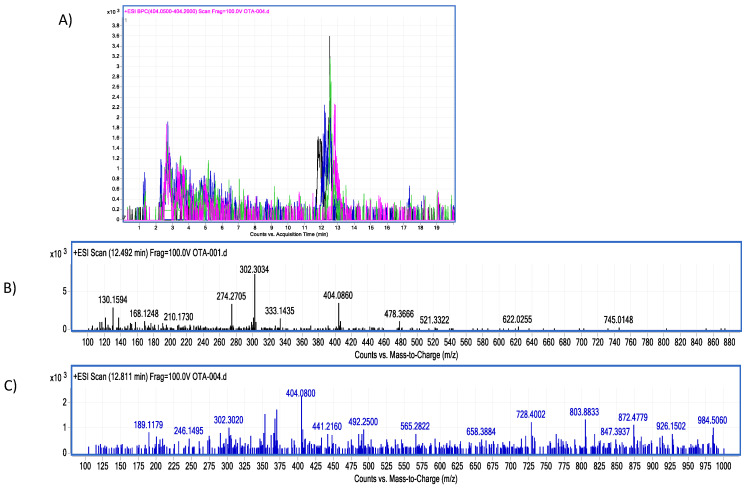
Bromelain–OTA biotransformation HPLC–TOF–MS chromatograms showing: (**A**) Ochratoxin (OTA) control (black), experimental assay and replicates (blue, green and pink). (**B**) OTA’s molecular ion as found in control tubes (M + 1) at 404.08 (*m*/*z*); (**C**) OTA’s molecular ion as found in tested tubes (M + 1) at 404.08 (*m*/*z*). The molecular ion (M + 1) of α-ochratoxin was not detected (fragmentation at 212.85 *m*/*z*).

**Figure 9 molecules-28-02019-f009:**
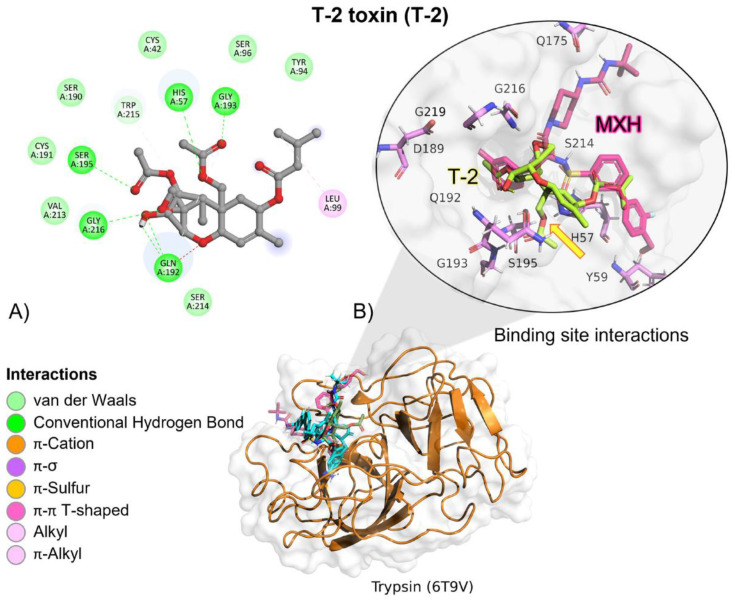
T-2 toxin–trypsin interactions. (**A**) Detailed view showing the interactions of the main different amino acid residues in trypsin with the T-2 toxin ligand. (**B**) Panoramic view of the reference (MXH) and the tested T-2 toxin ligands in trypsin. Comparison between the docked pose of the tested and the reference ligand within the bromelain binding pocket. The yellow arrow indicates the proximity between the trypsin catalytic triad and the T-2 toxin.

**Figure 10 molecules-28-02019-f010:**
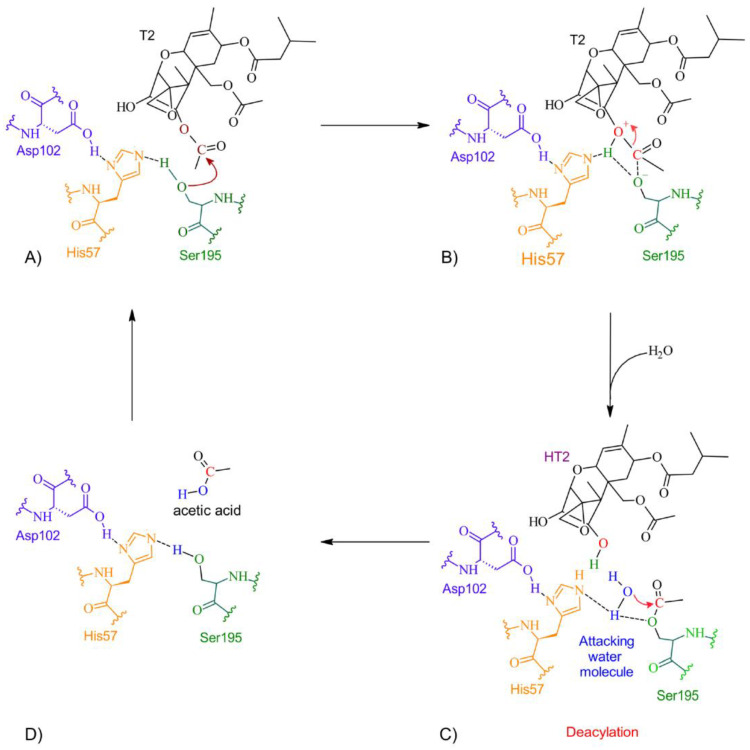
The detoxifying mechanism in T-2 by trypsin. (**A**) Ser195 attacks the carbonyl of the ester group in T-2, and an intermediate is formed. (**B**) The electron pair from the ester bond is transferred to the oxygen to break the ester bond. A proton is transferred from His57 to the hydroxide at the former molecule and HT-2 is released. (**C**) A water molecule enters and interacts with His57 and Ser195 to form an intermediate and the diacylation process initiates. The oxygen coming from the water molecule attacks the carbonyl with the Ser195. (**D**) A molecule of acetic acid is released, and the active site is restored.

**Figure 11 molecules-28-02019-f011:**
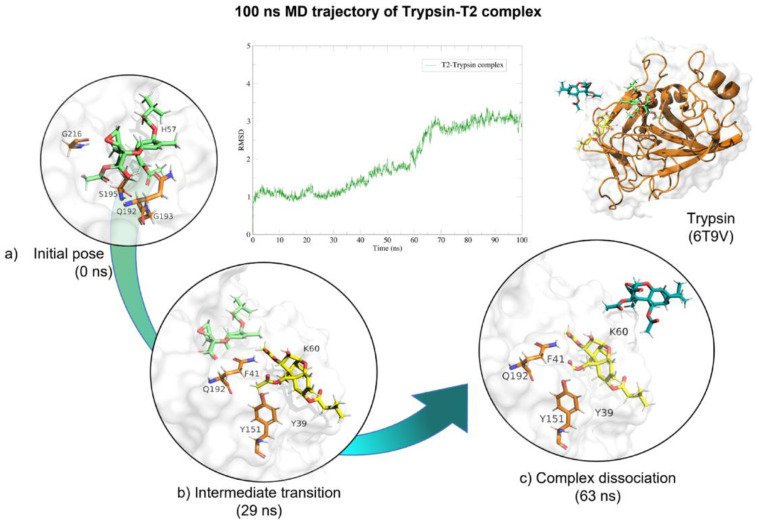
Molecular dynamics of trypsin–T2 complex. The root mean squared deviation (RMSD) plot shows changes in the complex conformation with respect to the initial state during 100 (ns). (**a**) Initial pose (0 ns), complex initial state obtained from molecular docking studies. (**b**) Intermediate transition pose visualization where T2 moves within the ligand binding site. (**c**) Complex dissociation, T2 becomes unbound from trypsin binding site. Green, T2′s initial pose; Yellow, T2 at 29 (ns); Dark green, T2 at 73 (ns).

**Figure 12 molecules-28-02019-f012:**
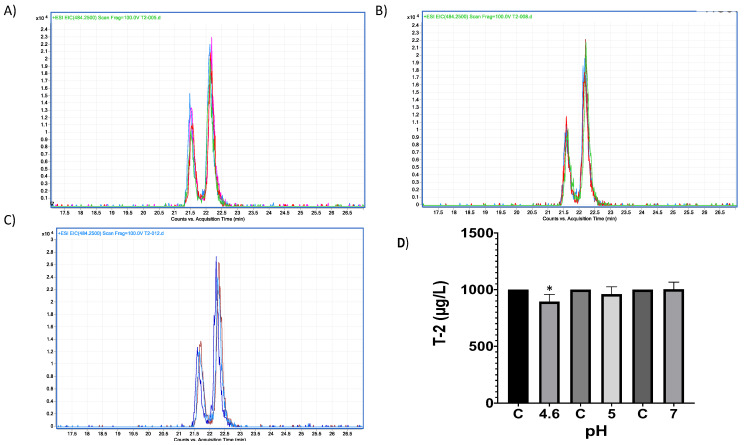
HPLC–TOF–MS chromatograms of trypsin-T2 in vitro experiments at different pH conditions and incubated a 41 °C. (**A**) pH 4.6; (**B**). pH 5; (**C**) pH 7. (**D**) Bars show T-2 (µg/L) degradation (10.68%) by trypsin at pH 4.6. Chromatograms include the control peak (red) and the experimental assay with two repetitions (pink, blue and green). The double peak belongs to T-2 (22.12 *m*/*z*). (* *p* < 0.05).

**Figure 13 molecules-28-02019-f013:**
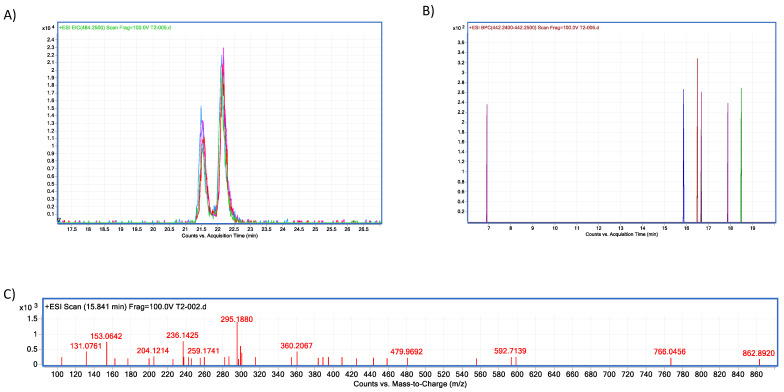
Trypsin–T2 biotransformation HPLC–TOF–MS chromatograms showing: (**A**) T-2 toxin control peak; (**B**) HT-2 molecular ion (M+1) final product of T-2 biotransformation by trypsin, as found in tested tubes; (**C**) T-2′s molecular ion as found in control tubes (M + 1).

**Figure 14 molecules-28-02019-f014:**
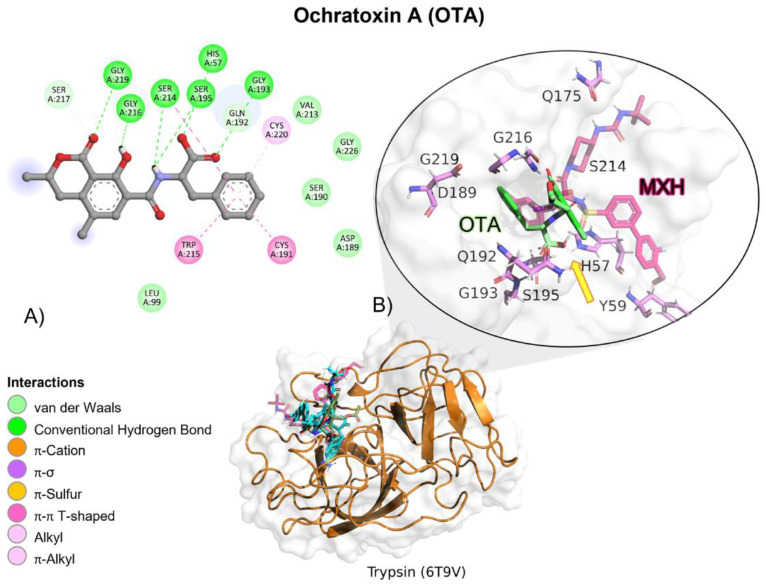
Ochratoxin (OTA)—trypsin interactions. (**A**) Panoramic view of the reference (MXH) and the tested OTA ligands within the trypsin binding site showing the docked poses of the tested and reference ligands. (**B**) Detailed view showing the interactions of the different amino acid residues within the binding site of trypsin with the OTA ligand. The yellow arrow shows the proximity between the trypsin catalytic triad and the OTA amide bond, suggesting its breakup.

**Figure 15 molecules-28-02019-f015:**
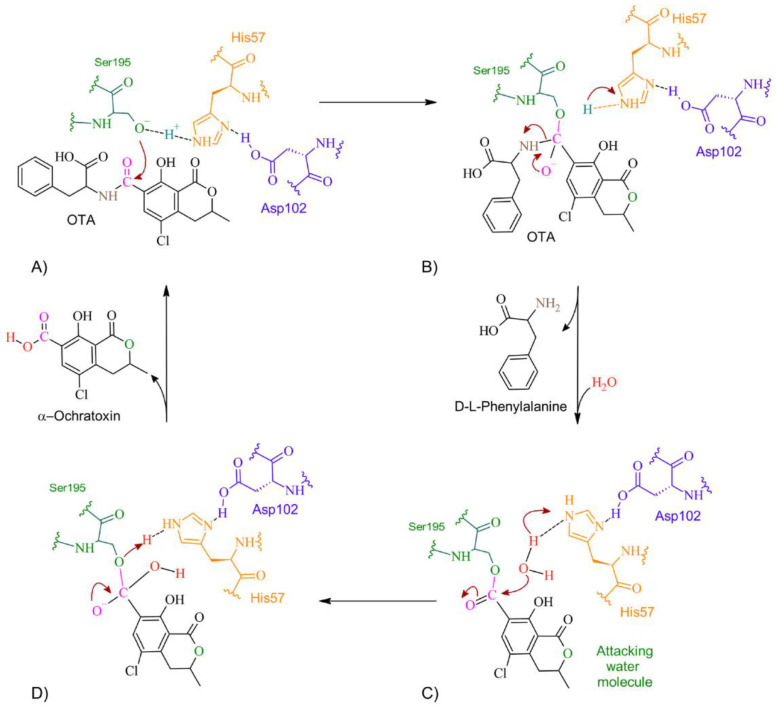
The detoxifying mechanism in OTA by trypsin. (**A**) Ser195 attacks the carbonyl of OTA’s peptide bond. (**B**) An intermediate is formed between Ser195 and the carbonyl group. There is an electron rearrangement. The carbonyl double bond with oxygen is restored and the peptide bond is broken. (**C**) D-L-phenylalanine is released, and a water molecule enters. His57 attacks the proton of water. The oxygen attacks the carbonyl, which shifts an electron pair from the double bond to the oxygen on the carbonyl. (**D**) The double bond of oxygen with carbon is re-established. The active site is regenerated by transferring a proton from His57 to Ser195. The molecule of α-ochratoxin is released.

**Figure 16 molecules-28-02019-f016:**
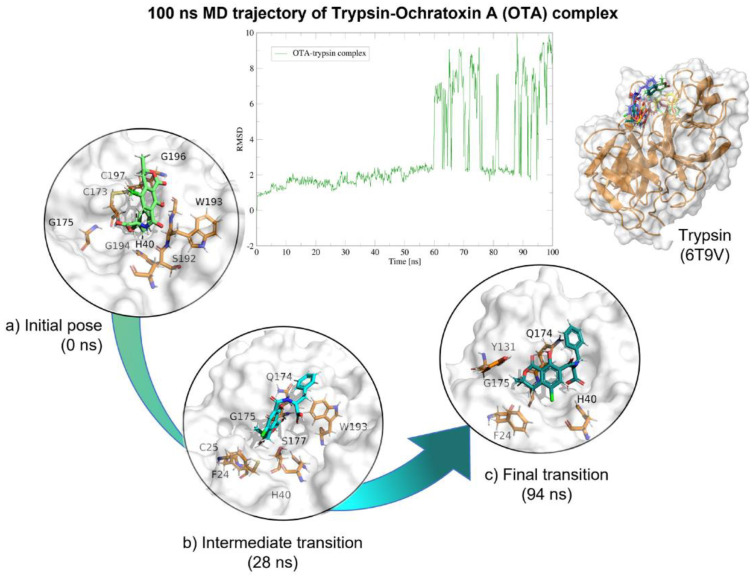
Molecular dynamics of the trypsin–OTA complex. The root mean squared deviation (RMSD) plot shows changes in the complex conformation with respect to the initial state during 100 (ns). The great variation in RMSD found from 58 (ns) is due to trypsin loops near the box limits throughout the 100 (ns). (**a**) Initial pose (0 ns), complex initial state obtained from molecular docking studies. (**b**) Intermediate transition pose visualization where OTA moves within the ligand binding site. (**c**) Final transition where OTA still remains at the trypsin binding site. Green, OTA’s initial pose; Blue, OTA at 28 (ns); Deep green, OTA at 94 (ns).

**Figure 17 molecules-28-02019-f017:**
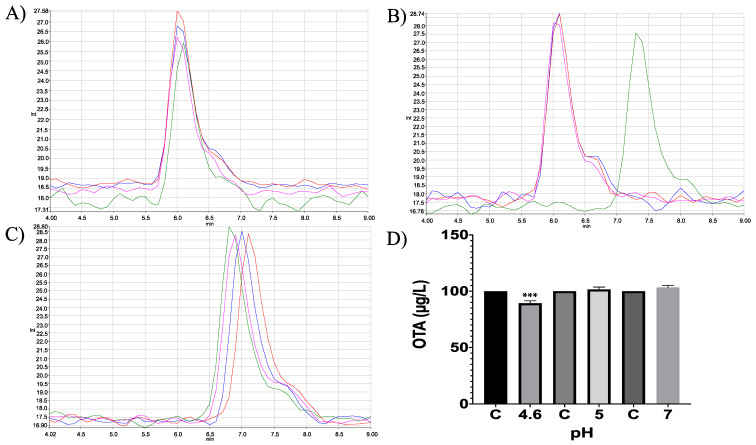
Trypsin in vitro experiments. HPLC-fluorescence chromatograms of trypsin–OTA in vitro assays showed differences in trypsin’s ability to degrade OTA at different pH conditions and incubated at 41 °C. (**A**) pH 4.6; (**B**). pH 5; (**C**) pH 7; (**D**) Bars show OTA (µg/L) degradation (10.69%) by trypsin at pH 4.6. In all chromatograms, the control is represented in red and experimental repetitions are in blue, green and pink. (*** *p* < 0.0001).

**Figure 18 molecules-28-02019-f018:**
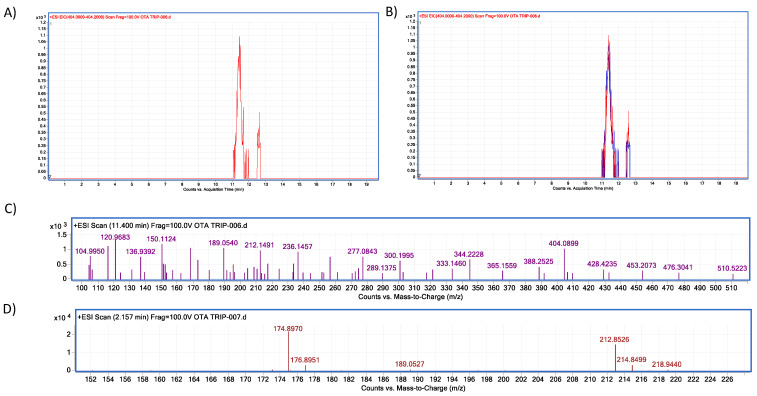
Trypsin–OTA biotransformation HPLC–TOF–MS chromatograms showing: (**A**) Ochratoxin (OTA) control peak; (**B**) Ochratoxin (OTA) peak in control (red) and experimental assays (blue); (**C**) OTA’s molecular ion as found in control tubes (M + 1) at 404.08 (*m*/*z*); (**D**) molecular ion (M + 1) of the most stable product of α–ochratoxin fragmentation at 212.85 (*m*/*z*).

**Figure 19 molecules-28-02019-f019:**
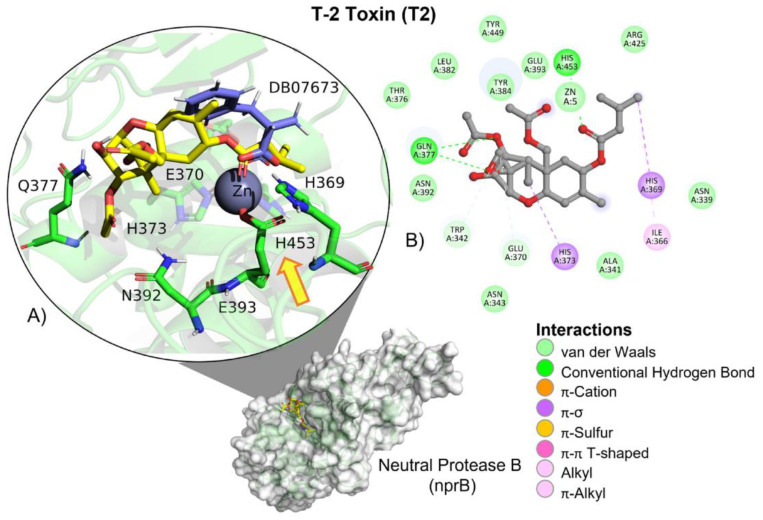
T-2 toxin–neutral metalloendopeptidase interactions. (**A**) Panoramic view of the reference (DB07673) and the tested T-2 toxin ligands within the binding site of neutral metalloendopeptidase. Comparison between the docked pose of both ligands within the binding site. The yellow arrow indicates the proximity between the neutral metalloendopeptidase catalytic triad and the T-2 toxin. (**B**) Interactions of the main different amino acid residues within the binding pocket of neutral metalloendopeptidase with the T-2 toxin ligand.

**Figure 20 molecules-28-02019-f020:**
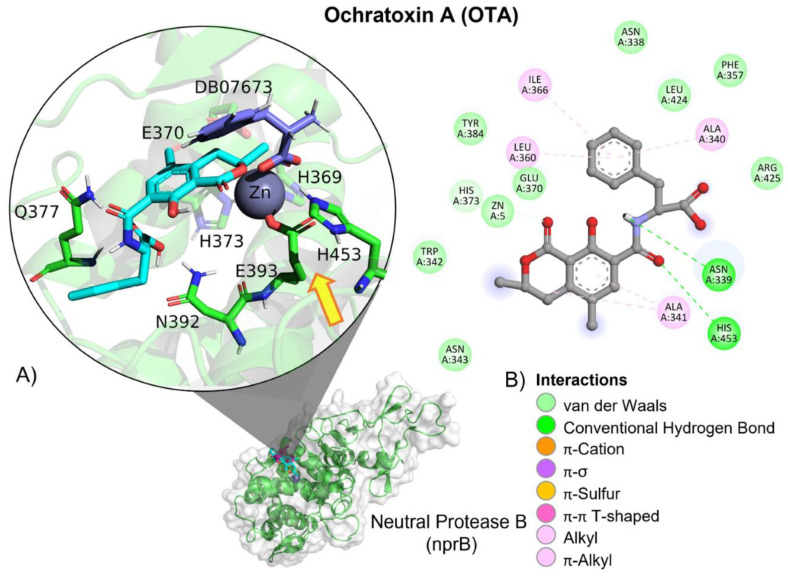
Ochratoxin (OTA)—neutral metalloendopeptidase interactions. (**A**) Panoramic view of the reference (DB07673) and the tested OTA ligands within a neutral metalloendopeptidase binding site, showing the docked poses of the tested and reference ligands. (**B**) Detailed view showing the interactions of the different amino acid residues within the binding site of neutral metalloendopeptidase with the OTA ligand. The yellow arrow shows the proximity between the neutral protease B catalytic triad and the OTA amide bond, suggesting its breakup.

**Figure 21 molecules-28-02019-f021:**
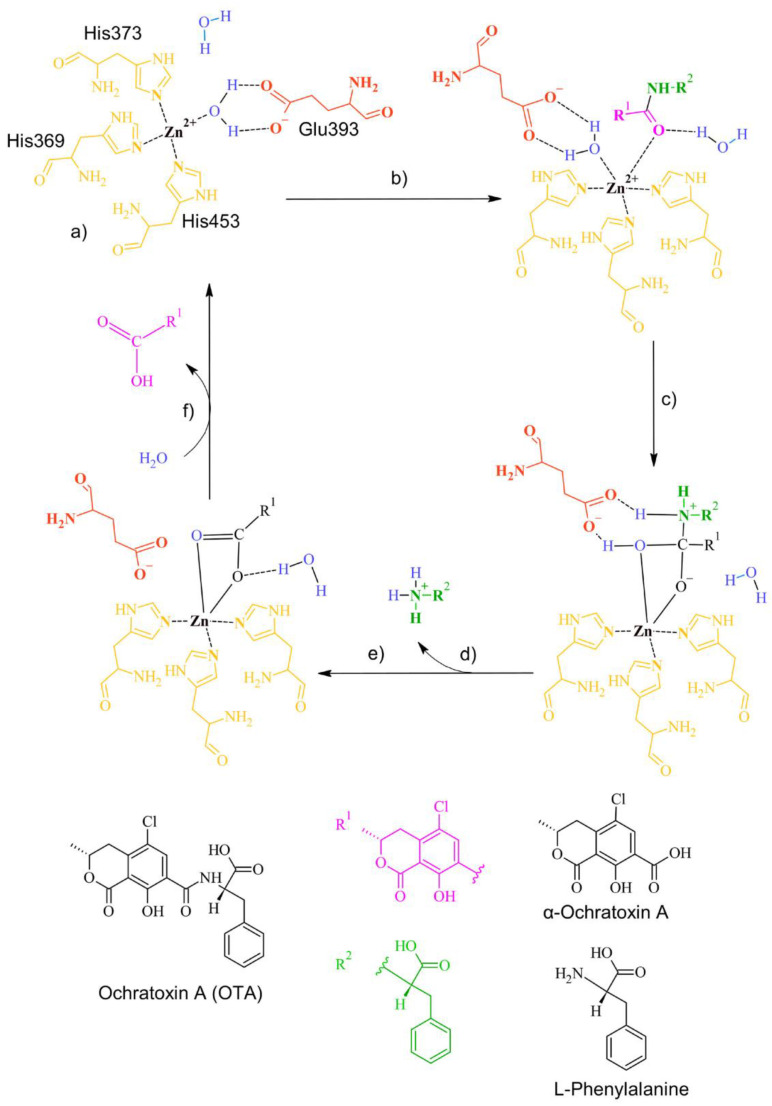
The detoxifying mechanism in OTA by neutral metalloendopeptidase. (**a**) Binding site of neutral metalloendopeptidase; (**b**) The carbonyl oxygen of OTA interacts with the Zn of the enzyme, forming a complex with a water molecule; (**c**) The complex between the water and the O of the carbonyl dissolves. A covalent bond between the O of the carbonyl and the Zn is formed. A complex between the O of the water molecule and the carbonyl of OTA is formed. A bond between the H of water and the amine group in OTA is formed; (**d**) The amine group is released; (**e**) The double bond of the carbonyl group with the O in OTA is restored. The interaction between glutamic acid and OTA is broken, and a complex between the water molecule and the O of OTA’s carbonyl is formed; (**f**) The carbonyl group is released, and the ingress of a water molecule restores the active site.

**Figure 22 molecules-28-02019-f022:**
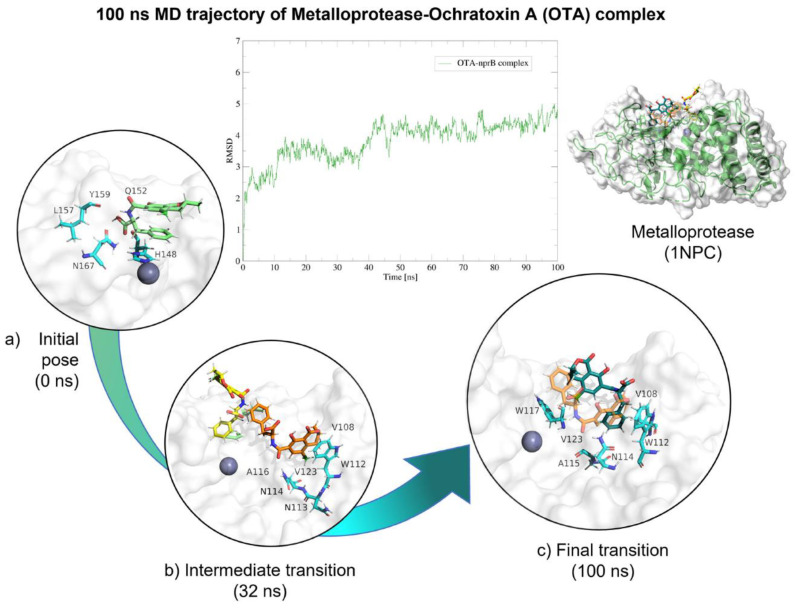
Molecular dynamics of metalloendopeptidase–OTA complex. The root mean squared deviation (RMSD) plot shows changes in the complex conformation with respect to the initial state during 100 (ns). (**a**) Initial pose (0 ns), complex initial state obtained from molecular docking studies. (**b**) Intermediate transition poses visualization where OTA moves from the ligand binding site. (**c**) Final transition showing no dissociation. Green, OTA’s initial pose; Yellow, OTA at 16 (ns); Orange, OTA at 30 (ns); Blue, OTA at 32 (ns), Dark green, OTA at 100 (ns).

**Figure 23 molecules-28-02019-f023:**
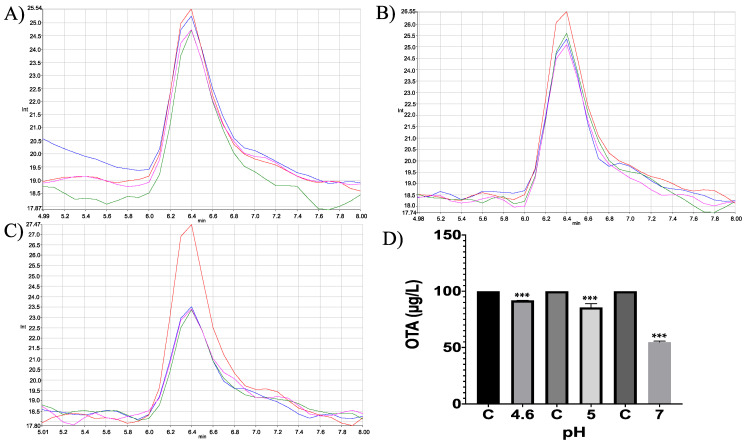
Neutral metalloendopeptidase in vitro experiments. HPLC-fluorescence chromatograms of neutral metalloendopeptidase–OTA in vitro experiments, showing differences in its ability to degrade OTA at different pH conditions and incubated at 41 °C. (**A**) pH 4.6; (**B**). pH 5; (**C**) pH 7; (**D**) Bars showed OTA (µg/L) degradation (8.2, 14.44 and 45.26%) at pH 4.6, 5 and 7, respectively. In all chromatograms the control is represented in red and experimental repetitions are in blue, green and pink. (*** *p* < 0.0001).

**Figure 24 molecules-28-02019-f024:**
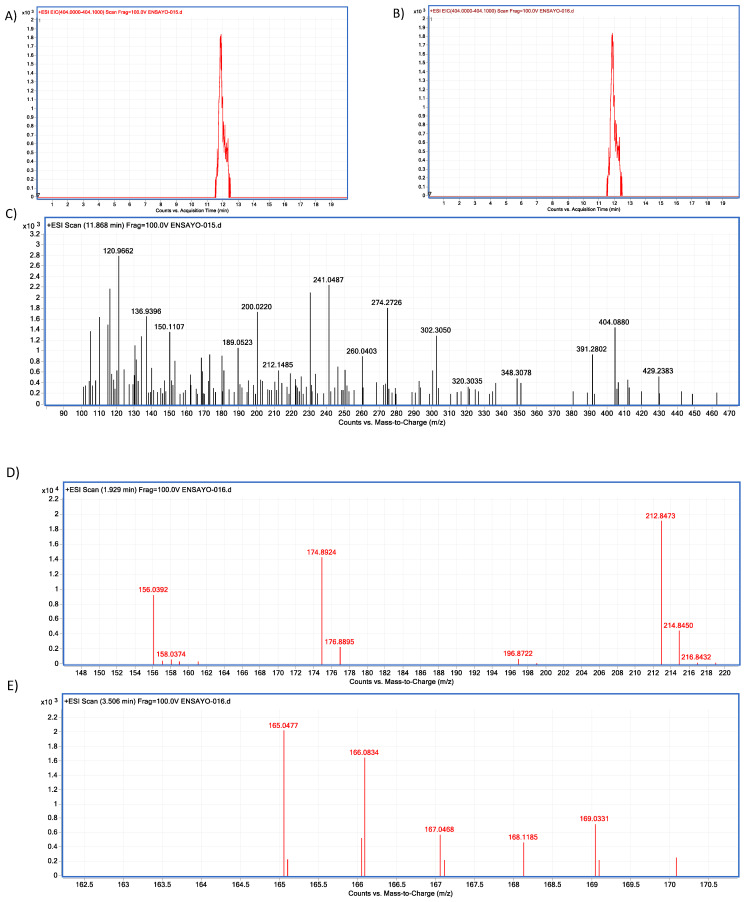
Neutral metalloendopeptidase–OTA biotransformation HPLC–TOF–MS chromatograms showing: (**A**) Ochratoxin (OTA) control peak; (**B**) Ochratoxin (OTA) peak is not present in tested tubes (red = control; brown line = tested tube); (**C**) OTA’s molecular ion as found in control tubes (M + 1) at 404.05 (*m*/*z*); (**D**) molecular ion (M + 1) of the most stable product of α-ochratoxin fragmentation at 212.84 (*m*/*z*); (**E**) Phenylalanine molecular ion (M + 1) at 166.08 (*m*/*z*) final product of OTA biotransformation by neutral metalloendopeptidase, as found in tested tubes.

**Table 1 molecules-28-02019-t001:** Chemical structure of tested and reference ligands.

ID	Name	2D Structure	Reference
T-2	T-2 toxin 4β,15-diacetoxy-3α-hydroxy-8α-(3-methylbutyryloxy)-12,13-epoxytrichotechec-9-ene,12,13-epoxytrichothec-9-ene-3,4,8,15-tetrol-4,15-diacetate-8-isovalerate	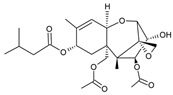	[54]
OTA	Ochratoxin A N-[(3R)-(5-chloro-8-hydroxy-3-methyl-1-oxo-7-isochromanyl) carbonyl]-L-phenylalanine	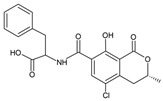	[23]
E64	N-[N-[1-hydroxycarboxyethyl-carbonyl] leucylamino-butyl]-guanidine	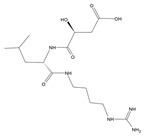	[47]
MXH	1-~{tert}-butyl-3-[1-[(2~{S})-3-(3-carbamimidoylphenyl)-2-[[3-[3-fluoranyl-4-(hydroxymethyl)phenyl]phenyl]sulfonylamino]propanoyl]piperidin-4-yl]urea	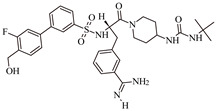	[55]
DB07673	(2S)-2-Methyl-3-phenylpropanoic acid	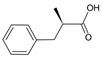	[56]

**Table 2 molecules-28-02019-t002:** Free binding energy and affinity (*K_i_*) of tested ligand–enzyme complexes were analysed with different scoring function software.

Protein	Ligand	Autodock Vina * 1.2.3	Swiss Dock *	Autodock 4.2.6	
		Free Binding Energy (kcal/mol)	Free Binding Energy (kcal/mol)	Free Binding Energy (kcal/mol)	*K_i_* (μM)
Bromelain cysteine	T-2	−6.06	−8.14	−5.45	101.78
	OTA	−5.94	−7.46	−5.63	74.14
	E64	−6.43	−7.76	−4.44	553.81
Bovine trypsin	T-2	−5.79	−6.72	−6.94	8.16
	OTA	−7.75	−7.72	−7.52	2.82
	MXH	−8.32	−8.21	−10.25	0.03059
Neutral metalloendopeptidase B	T-2	−6.45	−6.86	−6.21	28.06
	OTA	−7.57	−8.56	−5.94	44.05
	DB07673	−5.23	−9.35	−3.82	1.59

* software analyses did not provide a calculated *K_i_* value.

**Table 3 molecules-28-02019-t003:** Protein–ligand main interactions.

Protein	Ligand	Residues
Bromelain cysteine	T-2	H Bond: Gln20, Cys26, Asn157.
	OTA	H Bond: Gln24, Cys26, Gly66. Aliphatic: Ala136. π-π T shaped: His158, Trp181.
	E64	H Bond: Gln20, Cys26, His158, Gly66, Lys64. Aliphatic: Ala159, Ala133, Trp67.
Bovine trypsin	T-2	H bond: Ser195, His57, Gly193, Gly216, Gln192. Alkyl: Leu99 Van der Waals: Val213, Cys191, Ser190, Trp215, Cys42, Ser96, Tyr94, Ser2
	OTA	H Bond: Gly219, Gly216, Ser214, Ser195, His57, Gly193 Amide-π: Trp215, Cys191 π -alkyl: Cys220 Van der Waals: Ser217, Gln192, Val213, Gly226, Ser190, Asp189, Leu99
	MXH	H Bond: Asp189, Gly219, Gly216, Gln175. Aliphatic: Leu99 Amide-π: Cys191 Halogen: Asn97 Van der Waals: Thr98, Ser217, Ser195, Trp215, Val213, Ser190, Gly226, Cys220, Ser214, His57, Gln192.
Neutral metalloendopeptidase B	T-2	H Bond: His453, Gln377. Aliphatic: Ile366. π-σ: His369, His373.
	OTA	H Bond: Asn339, His453. Aliphatic: Ile340, Leu360, Ile366.
	DB07673	Attractive charge: Arg425, His453, Zn ion. π-σ: Ala341.

**Table 4 molecules-28-02019-t004:** Protein–ligand main poses in molecular dynamics.

Protein	Pose (ns)	Residues
Bromelain-OTA	0	H Bond: Gly65, Ser24, Cis25. π -alkyl: His158, Trp181, Ala132. Amide-π: Phe66, Asn157, Leu156.
	51	H Bond: Gly65, Trp26, Ala159. π -alkyl: Ala132, Leu156. Van der Waals: Gly65, Phe66, His158.
	53	Amide-π: Phe66. π -alkyl: Ala132. Van der Waals: Ser136, Asn157, His158, Ala159, Leu156, Ser155.
	100	π- π: Tyr86. Van der Waals: Ser99, Ala97, Asn98, Asp85.
Bovine trypsin-T2	0	H bond: Gln192, GGly216, Ser195, Hys57, Gly193. π -alkyl: Hys57. Alkyl: Leu99. Van der Waals: Gly219, His220, Val227, Ser190, Cys191, Val213, Ser214, Cys42, Phe41, Ser96, Tyr94.
	29	H bond: Lys60, Gln192. π -anion: Tyr39. Van der Waals: Hys57, Cys58, Phe41, Gly142, Gly193, Tyr151.
	63	H bond: Tyr39. Van der Waals: Lys60, Phe41, Ser37, Gly38.
Bovine trypsin-OTA	0	H bond: Gly175, His40, Ser192, Gly194, Gly196. π -alkyl: Cys197. Amide-π: Trp193, Cys173. Van der Waals: Ser172, Val191, Val205, Gly204, Asp171, Asp176, Ser177, Leu81.
	29	H Bond: Gly175. Amide-π: Gly194, Ser195. Van der Waals: Asp176, Trp193, Cys173, Ser172, Gly196, Tyr22, Ser177, His40, His23, Cys41. π -cation: Phe24. π -sulphur: Cys197. Halogen: Cys25.
	78	H Bond: Gln174, Tyr131, Ser177. π -sulphur: Cys197. Van der Waals: Gly175, Trp193, Ser192, Gly194, Cys173, Phe24, Cys25, Cys41, His40.
	90	H Bond: Gly175. π-σ: Trp193. Alkyl: Cys25. Van der Waals: Ser195, Tyr131, Phe24, His40.
	91	H Bond: His40 Van der Waals: Tyr131, Ser195, Trp193, Ser177, Gly175, Cys41, Cys25. π- donor hydrogen bond: Gln174. π-alkyl: Phe24.
Neutral metalloendopeptidase B-OTA	0	H Bond: Gln152, Leu157, Asn167, His148, Tyr159. Van der Waals: Asn118, Gly156, Gly119, Ala116, Glu155, Leu158, Trp 117, Tyr224.
	16	H Bond: Tyr159, Asn118. π- π: His148. Van der Waals: Gly119, Trp117, Glu168, Gly152, Asn167.
	32	H Bond: Asn114. π- π: Trp112. Alkyl: Val123, Val108. π-alkyl: Ala116. Van der Waals: Gly109, Gly1125, Tyr124, Ala115, Trp117, Asn118.
	100	π-alkyl: Val101, Val123. Van der Waals: Asn114, Ala116, Ala115, Tyr124, Gly125, Trp112, Trp117, Asn118.

## Data Availability

Not applicable.

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
