# Peer review of "Molecular Docking and In Vitro Studies of Ochratoxin A (OTA) Biodetoxification Testing Three Endopeptidases"

_molecules, 2023, doi:10.3390/molecules28052019_

Round 1

Reviewer 1 Report (Previous Reviewer 1)

The paper by Orozco-Cortés et al., which has been revised and resubmitted, shows that the degradation of ochratoxin A is too low (Figs. 6D, 10D, 14D, 19D) and, it is quite unreasonable to discusses the enzymatic degradation capacity and degradation mechanism. Therefore, I judge that it is completely unsuitable for publication in Molecules.

Author Response

Manuscript ID: molecules-1963135, entitled: "Molecular docking and in vitro studies of Ochratoxin A (OTA) biodetoxification using Ananas comosus bromelain cysteine-protease and Bacillus subtilis neutral metallo-protease".

We really appreciate the Associate editor and reviewers for their patience and dedication to follow-up this manuscript, particularly for your unvaluable observations to keep improving and making our manuscript more complete and clearer. In this latest version of the manuscript, we included all your requests, particularly related to: i) perform molecular dynamics; ii) improve figures quality; and iii) improve language.  Accordingly, we included equilibrium molecular dynamics (EMD) studies; redraw figures and detoxification mechanisms diagrams to provide a better visibility, and language was revised by an English native speaker.  As a result, the manuscript is now more complete and readable.

REVIEWER 1. Comment 1.  The paper by Orozco-Cortés et al., which has been revised and resubmitted, shows that the degradation of ochratoxin A is too low (Figs. 6D, 10D, 14D, 19D) and it is quite unreasonable to discusses the enzymatic degradation capacity and degradation mechanism. Therefore, I judge that it is completely unsuitable for publication in Molecules.  We really appreciate your comment and understand your concern regarding the enzymatic ability to break down mycotoxins.  We consider important to mention that since mycotoxins are not the natural substrates of the studied enzymes, and further considering that they are toxic compounds, any percentage that is achieved reducing its concentration in feed, by the way, could reduce poultry from becoming seriously ill, and therefore prevent affections in productivity.  However, future studies should focus on developing in vivo assays to clarify this concern.  Moreover, pH conditions tested in this study were chosen based on the optimum of the enzymes, in addition to considering the pH variation of poultry digestive track conditions.  We believe that contrasting the activity of different proteases tested at different pH conditions, is important to the test its effect on the enzymatic degradation capacity.  In fact, our results showed that a small change in pH, from 4.6 to 5 significantly contributes to a greater mycotoxin degradation, making the degradation process more efficient.

Reviewer 2 Report (Previous Reviewer 2)

The manuscript was modified according to previous comments and suggestions.

Author Response

Manuscript ID: molecules-1963135, entitled: "Molecular docking and in vitro studies of Ochratoxin A (OTA) biodetoxification using Ananas comosus bromelain cysteine-protease and Bacillus subtilis neutral metallo-protease".

We really appreciate the Associate editor and reviewers for their patience and dedication to follow-up this manuscript, particularly for your unvaluable observations to keep improving and making our manuscript more complete and clearer.  In this latest version of the manuscript, we included all your requests, particularly related to: i) perform molecular dynamics; ii) improve figures quality; and iii) improve language.  Accordingly, we included equilibrium molecular dynamics (EMD) studies; redraw figures and detoxification mechanisms diagrams to provide a better visibility, and language was revised by an English native speaker.  As a result, the manuscript is now more complete and readable.

REVIEWER 2. Comment 2.  The manuscript was modified according to previous comments and suggestions.  We greatly appreciate your patience and dedication in following-up this manuscript with your feedbackYour comments undoubtedly, were of great help for us to improve the latest version of this study.

I

Reviewer 3 Report (Previous Reviewer 3)

The authors responded and made the requested changes.

Author Response

Manuscript ID: molecules-1963135, entitled: "Molecular docking and in vitro studies of Ochratoxin A (OTA) biodetoxification using Ananas comosus bromelain cysteine-protease and Bacillus subtilis neutral metallo-protease".

We really appreciate the Associate editor and reviewers for their patience and dedication to follow-up this manuscript, particularly for your unvaluable observations to keep improving and making our manuscript more complete and clearer.  In this latest version of the manuscript, we included all your requests, particularly related to: i) perform molecular dynamics; ii) improve figures quality; and iii) improve language.  Accordingly, we included equilibrium molecular dynamics (EMD) studies; redraw figures and detoxification mechanisms diagrams to provide a better visibility, and language was revised by an English native speaker.  As a result, the manuscript is now more complete and readable.

REVIEWER 3. Comment 3.  The authors responded and made the requested changes.  We greatly appreciate your patience and dedication in following-up this manuscript with your feedbackYour comments undoubtedly, were of great help for us to improve the latest version of this study.

Reviewer 4 Report (New Reviewer)

The manuscript reports the Molecular docking and experimental studies of Ochratoxin A (OTA) biodetoxification testing three endopeptidases.

The manuscript is overall well organized and written. However, it requires some major revisions before being accepted for publication in this journal.

Minor comments:

Like We included docking experiments 19 with reference ligands and T-2 toxin as control, and in vitro bioassays.

Rephrase it, docking is not an experiment

There a few English - language revisions and minor spell check that I would recommend. Please also avoid using personal pronouns in formal scientific writing and therefore avoid expressions such as "Here we",  “To our” etc..

I would suggest the authors perform molecular dynamics studies.

The authors should justify the purpose of using different docking software in the manuscript.

Provide the proper visible Fig. 3, 4,. As similar to Fig 8.

Similarly, redraw the Scheme 5 of Figure 5, it must be of proper visual to the readers, similar to Fig 9.

Fig. 6D, provide the significant value, same in Fig. 10D, Fig.19D

Replace Ki by “Ki

Author Response

Manuscript ID: molecules-1963135, entitled: "Molecular docking and in vitro studies of Ochratoxin A (OTA) biodetoxification using Ananas comosus bromelain cysteine-protease and Bacillus subtilis neutral metallo-protease".

We really appreciate the Associate editor and reviewers for their patience and dedication to follow-up this manuscript, particularly for your unvaluable observations to keep improving and making our manuscript more complete and clearer.  In this latest version of the manuscript, we included all your requests, particularly related to: i) perform molecular dynamics; ii) improve figures quality; and iii) improve language.  Accordingly, we included equilibrium molecular dynamics (EMD) studies; redraw figures and detoxification mechanisms diagrams to provide a better visibility, and language was revised by an English native speaker.  As a result, the manuscript is now more complete and readable.

In the following sections we respond to each comment provided by the reviewer.  Each comment is listed and followed by our response. The remarks made by the Editor/Reviewer are written in bold face, and our respective answers are included in italics.

REVIEWER 4

Comment 4.  The manuscript reports the Molecular docking and experimental studies of Ochratoxin A (OTA) biodetoxification testing three endopeptidases.  The manuscript is overall well organized and written. However, it requires some major revisions before being accepted for publication in this journal.

Minor comments:

Comment 5.  Like We included docking experiments 19 with reference ligands and T-2 toxin as control, and in vitro bioassays.  Rephrase it, docking is not an experimentIn agreement with the reviewer, we paraphrased this sentence, here and throughout the manuscript, changing “experiments” by “studies”.

Comment 6.  There a few English - language revisions and minor spell check that I would recommend. Please also avoid using personal pronouns in formal scientific writing and therefore avoid expressions such as "Here we",  “To our” etc..  Language was checked by a native English speaker, and as requested, the whole manuscript was paraphrased avoiding personal writing.

Comment 7.  I would suggest the authors perform molecular dynamics studies.  Accordingly, molecular dynamics studies were performed and included in the latest version of the manuscript.

Comment 8.  The authors should justify the purpose of using different docking software in the manuscript.  We chose to perform docking analyses using three software differing in their scoring function to reduce biases and if similar results among them are obtained, reliability in the selection process increases and with more confidence experimental decisions can be taken optimizing costs.  Now it reads (current lines X) “ Analyses were performed in Autodock Vina [78, 79], Swiss Dock [80] and Autodock 4.2.6 [59] due to their differences in their molecular docking software’s scoring function, which reduces results biases and increase reliability.”

Comment 9.  Provide the proper visible Fig. 3, 4,. As similar to Fig 8.  Not only Figures 3, 4 and 8, but all figures were modified providing proper visibility of details by enhancing letter size and avoiding the use of dark backgrounds.

Comment 10.  Similarly, redraw the Scheme 5 of Figure 5, it must be of proper visual to the readers, similar to Fig 9.  Schemes were redrawn to provide proper visibility.

Comment 11.  Fig. 6D, provide the significant value, same in Fig. 10D, Fig.19D    As suggested, significant values were indicated in all bar figures and in figure legends.

Comment 12.  Replace Ki by “Ki  “Ki” was replaced by “Ki” throughout the manuscript.

Round 2

Reviewer 1 Report (Previous Reviewer 1)

The manuscript has been revised, so I think it is acceptable to publish it.

Reviewer 4 Report (New Reviewer)

The authors have removed the comments in their revised version.

Thanks

This manuscript is a resubmission of an earlier submission. The following is a list of the peer review reports and author responses from that submission.

Round 1

Reviewer 1 Report

In this paper, Orozco-Cortés et al. carried out the degradation experiments of ochratoxin A by enzymes, proteases. Development of a safe decomposition method for ochratoxin A is a very important issue. The degradation of ochratoxin A by proteases has been known for a long time, and unfortunately, the authors' current work has provided little new knowledge. Bromelain, which scored highly in the docking experiment, was actually unable to degrade ochratoxin A. On the other hand, similar metalloproteases could be degraded, making docking experiments meaningless. In this paper, only the result that Bacillus subtilis metalloprotease degraded ochratoxin A is reported. Therefore, I judge that it is not suitable for publication in Molecules.

Reviewer 2 Report

This is an interesting, well written manuscript describing the detoxification of ochratoxin A (OTA) by two commercial enzymes using molecular docking techniques and "in vitro" assays. The methodological approach seems correct and the results are acceptable, provided that the analytical method for residual OTA has been fully validated.

Specific comments:

-L.349 (also L.353): "ppb" is not an international standard unit.

-L.388: Use "x.g" instead of "rpm".

-L.361-368: Did you use calibration curves for OTA determinations? Was the analytical method validated? This is fundamental for accepting the results presented.

-L.377-378: Describe the "m/z" of precursor ion(s) and range examined.

Reviewer 3 Report

The manuscript entitled "Molecular docking and in vitro studies of Ochratoxin A (OTA) biodetoxification using Ananas comosus bromelain 2 cysteine-protease and Bacillus subtilis neutral metallo-protease" does not provide sufficient data to support mycotoxin degradation, in addition to data on the product generated from the possible degradation of the mycotoxin must be evaluated to verify the toxicity of the material.